# OpenReview forum: "Evolving Interpretable Constitutions for Multi-Agent Coordination"
_ICML.cc/2026/Conference — ICML 2026 regular_

### Official Review · Reviewer_VKp8 · 2026-03-11

**Soundness:** 3
**Presentation:** 4
**Significance:** 2
**Originality:** 3
**Overall Recommendation:** 5
**Confidence:** 5

**Summary:**

The authors propose the Constitutional Evolution framework, which uses OpenEvolve, an LLM-powered evolutionary framework, to evolve constitutions to govern multi-agent LLM behavior within a mixed-motive gridworld game focused on resource-gathering.
On this game, they demonstrate that OpenEvolve is able to discover constitutions that improve in social stability score by 67% over the best baseline (an LLM-generated constitution), as well as Anthropic’s “helpful, harmless, honest” constitution.

**Compliance With Llm Reviewing Policy:**

Affirmed.

**Final Justification:**

My final opinion of the paper remains similar to my original judgement, which is that the paper is well-executed, addresses and important problem, and is largely clearly written. The authors have addressed my comments satisfactorily.

**Key Questions For Authors:**

See main section of review.

**Limitations:**

yes, the authors clearly acknowledge limitations

**Strengths And Weaknesses:**

**Strengths**

- **Originality**: While LLM-powered evolutionary frameworks have been deployed for a variety of purposes, these frameworks largely result in code-base solutions. It’s interesting to see evolution successfully applied to generate “soft” guides to regulate multi-agent societal outcomes, and I think it’s an original application.

- **Well-motivated**: the problem of designing rules to regulate social behavior of LLMs in multi-agent  contexts is both important and well-motivated by the paper.

- **Clarity and presentation:** The paper is very clear and the analysis is sound. Most details are present, with a few exceptions (see below


**Weaknesses**

- **Only a single, toy-like task considered:** The method is only validated on a toy gridworld game. The lack of such a task hurts the significance of the paper. It’s not clear how well this would translate to more realistic scenarios / cooperative tasks such as reasoning or collaborative coding, or more commonly considered single-agent safety scenarios. Would the same approach transfer to broader scenarios? What would social stability look like in such a scenario? How would one define the equivalent of a social stability score there?

- **Lack of contextualization w.r.t. mechanism design**:  The paper studies the problem of  learning a set of rules that influence multi-agent interaction towards a more desirable outcome---this is essentially mechanism design (Sankar et al. 2026). However, the paper does not acknowledge the mechanism design literature. Within this literature, there is an approach that uses a similar LLM-powered evolutionary approach to discover mechanisms from data (Liu et al. 2025).

- **Unvalidated assumption that agents follow the constitution**: The paper relies on the assumption that agents will follow the provided constitution if prompted to do so. To contextualize the results, I think it would be useful to analyze how often constitutional violations occur under each constitution. As a follow-up, what happens to the  rule-violation rate if the scenario is seeded with a bad actor who does not follow the constitution, and acts antisocially?

- **Minor issues:** the prompt for OpenEvolve is not clearly mentioned in the appendix.


**Citations**

[1] Sankar, V. U., Rao, V. S., Bhardwaj, M. R., & Narahari, Y. (2026). *Deep Learning Meets Mechanism Design: Key Results and Some Novel Applications* (arXiv:2401.05683). arXiv. [https://doi.org/10.48550/arXiv.2401.05683](https://doi.org/10.48550/arXiv.2401.05683)

[2] Liu, J., Guo, M., & Conitzer, V. (2025). *An Interpretable Automated Mechanism Design Framework with Large Language Models* (arXiv:2502.12203). arXiv. [https://doi.org/10.48550/arXiv.2502.12203](https://doi.org/10.48550/arXiv.2502.12203)

---

> ### Author Rebuttal · Authors · 2026-03-31
>
> Thank you for your thoughtful and highly constructive review. We greatly appreciate your positive assessment of the paper’s originality, motivation, and clarity, and we are encouraged that you view the framework as a technically solid and interesting application of LLM-guided evolutionary search. Below we address the remaining concerns you raised.
>
> **[Task Scope / Generalization | Weaknesses]** “Only a single, toy-like task considered”
>
> Thank you for highlighting this limitation. We agree that the current evaluation in a 6×6 grid-world does not by itself establish transfer to more chaotic or realistic environments. This environment was chosen deliberately as a controlled, partially observable multi-agent setting with mixed incentives and survival pressure, allowing us to isolate the effect of constitutional rules and to interpret the resulting coordination dynamics clearly without confounding factors from more complex settings. We will revise the paper to make this scope more explicit.
>
> Regarding your specific questions on broader scenarios: The framework separates a task-agnostic search procedure from a task-specific welfare objective.
>
> - In our paper, the Stability Score combines "productivity", "survival", "conflict".
> - In broader domains, the same logic would be instantiated with domain-appropriate metrics. For example, in collaborative coding: "productivity" could reflect collective progress such as merged, test-passing code; "conflict" could capture harmful interference such as duplicated work, destructive edits, or unnecessary reversions; "survival" could reflect continued ability to contribute productively throughout the task. Therefore, extending the framework mainly requires instantiating the objective for the new domain rather than changing the core evolutionary search.
>
> **[Mechanism Design / Related Work | Weaknesses]** “Lack of contextualization w.r.t. mechanism design”
>
> Thank you for this excellent observation. We do briefly discuss mechanism design in our Social Welfare Theory subsection `Section 2.4`, where we note that "traditional mechanism design assumes known utility functions and game structures" and contrast this with our approach. However, we agree that the connection deserves fuller treatment, and we thank the reviewer for pointing us to Sankar et al. 2026 and Liu et al. 2025. We have expanded the discussion in two places:
>
> First, in the Evolutionary Search with LLMs subsection `Section 2.3`, we add Liu et al. 2025 as a concurrent example of LLM-powered evolution applied to rule discovery: "*Concurrently, Liu et al. (2025) reformulate mechanism design as a code generation task, using LLM-powered evolution to discover interpretable auction mechanisms. While their work targets economic settings and evolves code-level solutions, our framework evolves natural-language behavioral norms for multi-agent coordination.*"
>
> Second, in the Social Welfare Theory subsection `Section 2.4`, we expand our existing mechanism design discussion to cite the broader literature: "*Sankar et al. (2026) survey deep learning approaches to mechanism design, where neural networks learn mechanisms that approximately satisfy properties such as incentive compatibility and welfare maximization. Our work shares the goal of automated rule discovery for strategic agents, but differs in that traditional mechanism design typically assumes known utility functions and well-defined game structures, whereas we discover governance rules through evolutionary search over partially observable environments with LLM agents whose utility functions are implicit.*"
>
> **[Constitution-Following Assumption / Rule Violations | Weaknesses]** “Unvalidated assumption that agents follow the constitution”
>
> Thank you for your constructive comment. We agree that this is an important limitation. In the current paper, we evaluate constitutions at the outcome level —stability, productivity, conflict, behavior profiles, and reasoning traces—under the prompting assumption that agents attempt to follow the shared constitution. While the manuscript includes behavioral analysis and reasoning traces that help interpret how constitutions shape decisions, it does not yet introduce an explicit rule-violation metric or evaluate robustness to non-compliant agents.
>
> We will make this assumption more explicit in the revision and highlight two important next steps:
> - a clear metric for the direct auditing of constitutional compliance,
> - and robustness tests in settings with non-compliant or adversarial agents.
>
> We agree that such analyses would strengthen the framework and help clarify when prompt-based constitutions are sufficient versus when stronger enforcement mechanisms may be needed.
>
> **[OpenEvolve Prompt / Appendix Clarity | Minor Issue]** “the prompt for OpenEvolve is not clearly mentioned in the appendix”
>
> Thank you for pointing this out. The full mutation prompt is provided in `Section H.3` of the appendix. We will label it more clearly.

---

> > ### Author Rebuttal · Reviewer_VKp8 · 2026-04-03
> >
> > I thank the authors for their response, and encourage them to update the paper accordingly. I maintain my score and recommendation for acceptance.

---

> > > ### Author Response · Authors · 2026-04-06
> > >
> > > We thank the reviewer for the constructive and professional dialogue. We will incorporate the suggested clarifications in the final revision.

---

### Official Review · Reviewer_My3a · 2026-03-12

**Soundness:** 3
**Presentation:** 2
**Significance:** 2
**Originality:** 2
**Overall Recommendation:** 2
**Confidence:** 5

**Summary:**

This manuscript explores the limitations of traditional, static principles created by humans for aligning multi-agent AI systems. It introduces a new framework called Constitutional Evolution. Using a multi-island evolutionary approach, the authors optimize a set of natural-language behavioral rules, referred to as a constitution, to improve coordination among multiple large language model (LLM) agents in a simulated grid-world environment. The resulting constitution significantly outperforms both human-designed baselines and one-shot LLM-generated baselines. It enhances societal stability, boosts productivity, reduces conflicts, and minimizes unnecessary communication. The findings underscore the advantages of automated optimization for multi-agent alignment, highlighting key aspects such as interpretability, implicit coordination, and the potential for developing scalable, context-specific social contracts.

**Compliance With Llm Reviewing Policy:**

Affirmed.

**Final Justification:**

I thank the authors for their detailed rebuttal. Although the responses are helpful to some extent, they do not fully address my core concerns.

The manuscript proposes a framework for large model-guided agent learning and demonstrates its effectiveness in a small grid-world environment. However, the authors acknowledge in their response that they do not claim the framework can generalize to larger or more complex environments. This significantly undermines the approach's scalability and raises serious doubts about its potential value to the multi-agent community. If further experimental validation is not feasible, it would be difficult to consider the framework suitable for publication in ICML. Therefore, I see no reason to revise my original assessment.

**Key Questions For Authors:**

Can the algorithm be transferred to larger environments with more agents? How effective is it across different environments?

**Limitations:**

The discussion of potential negative societal impacts is very limited.

**Strengths And Weaknesses:**

The manuscript introduces a framework, Constitutional Evolution, that develops natural-language behavioral rules to enhance coordination among multiple LLM agents. This topic is highly relevant to ongoing research in multi-agent alignment and AI governance. The concept of dynamically optimizing constitutions, rather than relying on static, human-designed rules, is particularly intriguing. The experiments indicate that the evolved constitution can enhance system stability and productivity compared to several baseline models. However, the experimental setup is somewhat limited, and a few presentation and evaluation issues diminish the work's overall impact.
Soundness: The proposed framework for evolving natural-language constitutions through a multi-island evolutionary strategy is conceptually sound. Experimental results provide preliminary evidence that this approach can outperform both human-designed and one-shot LLM-generated constitutions. However, the experiments are primarily conducted in a small 6×6 grid-world environment, and it remains unclear how well the method scales to larger or more complex settings. Additionally, some symbols in Equation (2) are not sufficiently defined.

Presentation: The manuscript is generally readable, and the motivation behind the work is clear. However, the overall formatting appears somewhat unrefined, with noticeable whitespace in certain sections. There are also formatting issues with some figures; for instance, the font size in Figure 1 is disproportionately large compared to the main text, and the notation for C∗ is inconsistently used.

Significance: The concept of evolving interpretable constitutions for multi-agent coordination holds potential significance for future research on AI alignment and governance. However, due to the limited experimental validation and the simplicity of the environment, the practical impact of the current work remains restricted.

Originality: This work offers an intriguing perspective by applying evolutionary optimization to the design of natural-language constitutions for multi-agent systems. Nevertheless, the method primarily combines existing components rather than introducing fundamentally new algorithms or theoretical insights, resulting in relatively moderate methodological novelty.

Other comments:

(1)	The overall layout of the manuscript is not sufficiently polished, and the formatting appears suboptimal. Several pages contain noticeable excessive whitespace. For example, there is a large blank area above Section 4.5, which affects visual coherence and reading continuity.

(2)	The font size in Figure 1 is excessively large and exceeds that of the main text, resulting in a disproportionate visual presentation. Moreover, the figure conveys limited substantive information relative to the space it occupies. It is recommended to reduce the font size and refine the layout accordingly. A similar issue is observed in Figure 4 and should be addressed as well.

(3)	Several symbols appearing in Equation (2) are not adequately defined. The authors should provide explicit explanations for each symbol to improve clarity and rigor. In addition, the notation for C* is inconsistent between the equation and the main text; the symbol usage should be unified to avoid potential confusion.

(4)	Appropriate references should be provided when introducing the baseline algorithms to ensure proper attribution and academic rigor. At present, Baseline (1) lacks a citation, and the corresponding reference should be included to clarify its source.

(5)	The evaluation metrics reported in the experimental tables are not sufficiently explained. The authors should provide clear definitions and descriptions for each metric to enhance clarity and rigor. Moreover, in Table 6, under the S(X.3) metric, the result that is identical to that of C∗ is not highlighted in bold. In Table 8, under the Idle metric, the value corresponding to C∗ is bolded even though it is neither the highest nor the lowest result. The criteria for bold formatting appear inconsistent and should be clarified and applied uniformly across all tables.

(6)	The current experimental environment is a 6×6 grid world, and the figure results are a little. Furthermore, the algorithm may not effectively transfer to larger environments with more agents, and its effectiveness across different environments has not been thoroughly validated.

---

> ### Author Rebuttal · Authors · 2026-03-31
>
> Thank you for your detailed and constructive feedback. We appreciate your recognition that the problem is relevant and that the framework is conceptually sound. We would like to address your remaining concerns:
>
> **[Scalability / Generalization | Weaknesses]** “the current experimental environment is a 6×6 grid world … may not effectively transfer to larger environments”
>
> Thank you for raising this important point. We agree that the current evaluation in a 6×6 grid-world does not by itself establish transfer to larger or more realistic environments. This environment was chosen deliberately as a controlled, partially observable multi-agent setting with mixed incentives and survival pressure, allowing us to isolate the effect of constitutional rules and to interpret the resulting coordination dynamics clearly without confounding factors from more complex settings. We will revise paper, especially "introduction" `section 1` to make this scope more explicit.
>
> At the same time, we note that the proposed framework is not intrinsically tied to a specific map size or agent count, since it optimizes natural-language constitutions from simulation feedback rather than learning environment-specific model weights. Broader validation across larger populations, different map sizes, and more diverse task domains remains an important direction for future work.
>
> **[Formatting / Figures / Notation  | Weaknesses]**
>
> Thank you for these careful observations. We agree that the initial submission was not polished enough visually. In the revision:
> 1. We will clean up the whitespace and layout, including the blank area near Section 4.5;
> 2. We will revise Figures 1 and 4 so that font sizes are proportionate to the main text and space is used more efficiently. Please find them from here: (https://anonymous.4open.science/r/figures-rebuttal/)
> 3. we will standardize the notation for C* throughout the paper.
>
> Regarding `Equation (2)`, In the revision, we will restate the symbols directly around the equation and clarify that the objective is to maximize expected Stability Score over trajectory distribution induced by a constitution.
>
> **[Baselines / References | Weaknesses]**
> “Appropriate references should be provided when introducing the baseline algorithms”
>
> Thank you for pointing this out. We would like to clarify that Zero-sum baseline is a task-specific adversarial constitution introduced as a stress-test baseline rather than a canonical algorithm from prior work. To make it clear: In `section 4.1.3`, we will revise the description to make this role explicit. In `Appendix B.1`, we will also clarify that this baseline is included to verify that the environment permits harmful behavior, that evolutionary search can escape adversarial local optima, and that the Stability Score penalizes destructive conflict as intended.
>
> **[Metrics / Tables | Weaknesses]** “The evaluation metrics … are not sufficiently explained”
>
> Thank you for the careful observation. We will make both the metric definitions and the table-formatting convention clearer. In the revision, we will define all reported metrics explicitly before the results table and apply a consistent bolding convention (best result per column) across all tables. This will also resolve the specific inconsistencies you pointed out in Table 6 and Table 8.
>
> **[Originality | Weaknesses]** “the method primarily combines existing components”
>
> We appreciate this observation. We acknowledge that the individual components of our framework: evolutionary search, LLM-based mutation, multi-agent simulation, are established techniques. The contribution of this work, instead, is viewed as a novel synthesis targeted at a different alignment problem: optimizing natural-language constitutions for multi-agent coordination, rather than evolving programs/algorithms or designing constitutions by hand.
>
> To avoid any confusion, we will add novelty clarification to the `section 1`.
>
> **[Societal Impact | Limitations]** “The discussion of potential negative societal impacts is very limited”
>
> Thank you for raising this important point. We agree that the initial discussion of negative societal impacts was insufficient. We will expand our Limitations section `Appendix J`:
>
> *Broadly, discovering governance rules through evolutionary optimization carries inherent societal risks. Optimizing for a scalar stability metric introduces the risk of specification gaming, where an evolutionary process might discover and scale coercive, deceptive, or exclusionary behaviors that maximize the score but violate human ethical norms. Furthermore, constitutions evolved in simplified or idealized settings may encode assumptions that fail or cause harm when deployed in more complex real-world systems. Deploying such frameworks in practice requires rigorous secondary qualitative audits to prevent the propagation of harmful emergent social dynamics, and human oversight of evolved rules remains essential before any real-world application.*

---

> > ### Author Rebuttal · Reviewer_My3a · 2026-04-04
> >
> > I thank the authors for their detailed rebuttal. While the responses are helpful to some extent, they do not fully address my core concerns. Although the authors provide explanations regarding the limitations of the experimental setup, the lack of empirical evaluation in larger-scale environments remains a fundamental issue. Results obtained solely in a 6x6 environment are insufficient to support its effectiveness or potential applicability. In the absence of empirical evidence demonstrating scalability, the current results are not sufficiently convincing to establish that the proposed approach generalizes to more complex problem settings. Therefore, I consider this issue to significantly undermine the paper's overall persuasiveness.
> >
> > In addition, several issues remain that affect the overall presentation quality and the persuasiveness of the experimental results. Although Figures 1 and 4 have been revised, several concerns remain. Figure 1 appears overly simplified and resembles a basic flow diagram rather than a well-structured framework illustration, limiting its ability to convey meaningful information about the proposed method; Visual elements such as line thickness (red and black lines) and arrow sizes are inconsistent, which affects readability; The semantics of the arrows are not clearly explained, making it difficult for readers to interpret the process accurately.
> >
> > The updated version of Figure 4 also raises several questions. The previously included “LLM-Generated (S=0.332)” curve has been removed without explanation; Other baseline methods appear as nearly flat curves, showing little to no variation during training, which is atypical for reinforcement learning experiments. Further clarification or justification would be necessary.
> >
> > Aside from Figure 4, most experimental results are presented in tabular form, with no performance curves that reflect training dynamics. In multi-agent research, such curves are important for evaluating convergence behavior, stability, and learning efficiency. Relying primarily on tabular results weakens the overall credibility and completeness of the empirical evaluation.
> >
> > Overall, these issues reduce the clarity of presentation and the strength of empirical support. Therefore, I am not able to revise my score based on the current version.

---

> > > ### Author Response · Authors · 2026-04-06
> > >
> > > Thank you reviewer for kind and constructive follow-up. We would like to address your remaining concerns as follows:
> > >
> > > **[1] "Although the authors provide explanations regarding the limitations of the experimental setup, the lack of empirical evaluation in larger-scale environments remains a fundamental issue..."**
> > >
> > > We deeply respect the reviewer’s empirical standards. We agree that the lack of empirical evaluation beyond the 6x6 setting remains the limitation of the current submission, and we do not claim that the present paper establishes scalability to larger environments.
> > >
> > > However, our goal here is narrower: to provide controlled evidence, in a deliberately simplified and interpretable multi-agent setting, that iterative constitutional search can discover non-obvious coordination rules and can avoid failure modes such as the communication trap. In larger and more complex environments, the link between constitutional changes and system-level behavioral shifts may become harder to interpret. The 6x6 grid was therefore chosen to better isolate constitutional effects and support clear analysis of the resulting coordination dynamics.
> > >
> > > The paper already frames scale and transfer as open limitations, and we will make this even more explicit in the revision.
> > >
> > > **[2] "...In addition, several issues remain that affect the overall presentation quality... Figure 1 appears overly simplified and resembles a basic flow diagram rather than a well-structured framework illustration..."**
> > >
> > > Thank you reviewer for this constructive comment. We have revised Figure 1. Please find it from link below: https://anonymous.4open.science/r/figure-new-rebuttal/Figure_1_new.pdf
> > >
> > > In the revision, we standardized all line thicknesses and arrow styles; added explicit text annotations to every arrow explaining its exact semantic function (e.g., evaluate $C_n$ in simulation, return evaluation output, etc.); showed the current constitution $C_n$ as an actual rule snippet rather than an abstract placeholder, etc. These changes are intended to make the framework semantics more explicit and the figure more informative.
> > >
> > > **[3] "...The updated version of Figure 4 also raises several questions. The previously included “LLM-Generated (S=0.332)” curve has been removed without explanation;..."**
> > >
> > > Thank you reviewer for the careful observation and kind comment. We deeply apologize for this oversight. The removal of the “LLM-generated (S=0.332)” curve in the revised Figure 4 was a rendering error during our rebuttal formatting process. We have restored this curve to Figure 4, corrected the baseline math in the caption, and ensured all lines are accurately represented alongside their new variance bands.
> > >
> > > **[4] "...."Other baseline methods appear as nearly flat curves, showing little to no variation during training, which is atypical for reinforcement learning experiments..."**
> > >
> > > We apologise if our manuscript did not make this distinction clear enough. Our baselines are not reinforcement-learning agents. The baselines (HHH, Zero-Sum and the one-shot LLM-generated rules) are static, fixed text rulesets rather than iteratively optimized policies. Therefore, they are plotted in Figure 4 as flat horizontal reference lines showing their mean Stability Score across repeated evaluation runs. Our proposed method is the only one that evolves over iterations, which is why it is the only curve that changes over time.
> > > Please find the revised version of Figure 4 from here:
> > > https://anonymous.4open.science/r/figure-new-rebuttal/Figure_4_fix.pdf
> > >
> > > **[5] "...Aside from Figure 4, most experimental results are presented in tabular form, with no performance curves that reflect training dynamics. In multi-agent research, such curves are important for evaluating convergence behavior, stability, and learning efficiency..."**
> > >
> > > Thank you for this constructive comment. In our setting, only the proposed evolutionary search has optimization dynamics; the baselines are fixed constitutions and therefore do not have training curves in the RL sense. Figure 4 is intended to show the optimization trajectory of the evolving constitution, while the statistical robustness and single- vs multi-island comparisons provide complementary evidence on variance, convergence behavior, and stability.

---

### Official Review · Reviewer_mYqt · 2026-03-12

**Soundness:** 4
**Presentation:** 3
**Significance:** 3
**Originality:** 3
**Overall Recommendation:** 5
**Confidence:** 3

**Summary:**

This paper introduces Constitutional Evolution, a novel framework designed to automate the discovery of behavioral norms in multi-agent LLM systems. While previous "Constitutional AI" research primarily focused on aligning individual models with fixed principles, this work tackles the emergent social challenges—such as resource competition and conflict—that arise when multiple agents interact. The authors develop a multi-island evolutionary strategy where an LLM acts as a "mutation operator" to iteratively refine a set of natural language principles (the "Constitution"). These are evaluated based on a Societal Stability Score within a survival-pressure grid-world simulation. The framework successfully evolves specific, interpretable rules (e.g., "prioritize depositing over broadcasting") that significantly outperform standard alignment heuristics in maintaining social order and productivity.

**Compliance With Llm Reviewing Policy:**

Affirmed.

**Key Questions For Authors:**

NO questions.

**Limitations:**

The authors have adequately discussed the limitations, specifically pointing out the computational overhead of running multiple LLM-based simulation islands and the potential for "specification gaming," where agents might maximize the stability score through behaviors that the constitution did not explicitly forbid but humans might find undesirable.

**Strengths And Weaknesses:**

The submission is technically sound, employing a robust multi-island evolutionary strategy that effectively navigates the complex space of natural language rules while avoiding local optima through periodic "migration" and diverse mutation. Its presentation is highly commendable; the paper is logically structured, and the inclusion of qualitative "thought traces" in the appendix provides an excellent window into how the evolved constitution fundamentally reshapes agent reasoning. In terms of significance, the work marks a vital shift from individual model alignment to "social alignment," offering a scalable method to generate human-readable governance for future autonomous agent societies. Finally, the originality of the work is outstanding, as it provides a creative and well-articulated synthesis of evolutionary computation and LLM-driven agentic planning, successfully moving beyond static prompting into a dynamic, fitness-based paradigm of rule discovery. However, the work’s reliance on a simplified grid-world simulation raises questions about its scalability to more chaotic real-world environments, and the high computational cost of "LLM-in-the-loop" evolution remains a practical barrier for broader adoption.

---

> ### Author Rebuttal · Authors · 2026-03-30
>
> Thank you for your highly positive and thoughtful assessment. We greatly appreciate your recognition of the paper’s central contribution: shifting from individual-model alignment toward social alignment in multi-agent settings. We would like to address the limitations you highlighted as follows:
>
> **[Scalability / Scope | Limitations]** “the work’s reliance on a simplified grid-world simulation raises questions about its scalability”
>
> Thank you for raising this important point. We agree that the current evaluation in a 6×6 grid-world does not by itself establish transfer to larger or more realistic environments. This environment was chosen deliberately as a controlled, partially observable multi-agent setting with mixed incentives and survival pressure, allowing us to isolate the effect of constitutional rules and to interpret the resulting coordination dynamics clearly without confounding factors from more complex settings. We will revise the paper (especially "introduction" section) to make this scope more explicit and to avoid overstating generalization.
>
> At the same time, we note that the proposed framework is not intrinsically tied to a specific map size or agent count, since it optimizes natural-language constitutions from simulation feedback rather than learning environment-specific model weights. Broader validation across larger populations, different map sizes, and more diverse task domains remains an important direction for future work.
>
> **[Computational Cost | Limitations]** “the high computational cost of ‘LLM-in-the-loop’ evolution remains a practical barrier for broader adoption”
>
> We also thank the reviewer for highlighting the computational cost of “LLM-in-the-loop” evolution as a barrier to broader adoption. We agree that the current method is not yet optimized for large-scale deployment. At the same time, because the output of the method is an explicit natural-language constitution rather than updated model weights, the framework separates an expensive search phase from downstream applications of the discovered rules.
>
> **[Specification Gaming / Auditing | Limitations]**
> “the potential for specification gaming”
>
> Thank you for raising this important point. In fact, any optimization-based alignment framework may discover behaviors that score well under the chosen objective while still being undesirable under broader human judgment. One reason we emphasize interpretable constitutions is precisely that they make such behaviors inspectable: unlike opaque learned policies, the resulting rules can be directly read, and revised by humans.
>
> We thank you again for the strong support and careful reading. Your feedback reinforces the central framing of the paper: this work is intended as a first controlled demonstration that interpretable governance rules for multi-agent LLM systems can be discovered through evolutionary search, rather than only prescribed by hand. We will revise the manuscript to make the scope, limitations, and future directions even clearer.

---

> > ### Author Rebuttal · Reviewer_mYqt · 2026-04-04
> >
> > Thank you for your reply. I have no further questions.

---

> > > ### Author Response · Authors · 2026-04-06
> > >
> > > We thank the reviewer for the constructive and professional dialogue. We will incorporate the suggested clarifications in the final revision.

---

### Decision · Program_Chairs · 2026-04-30

**Decision:**

Accept (regular)

**Comment:**

This paper got very polarized reviews, both very positive and very negative. The strongest criticism against the paper seems to be that evaluation on a 6x6 grid is insufficient. I disagree with this view. Highly original ideas often require some initial "leap of faith" that they are, indeed, good ideas and it is nearly impossible to evaluate them fully and in as convincing settings as used for more traditional approaches.

I believe that the fact that this paper raises such polarized ideas---in this case---is an argument for accepting. It will raise interesting discussions at the conference and may, potentially, lead to important follow-up works.